# Effectiveness of Simulation with a Standardized Patient on Knowledge Acquisition, Knowledge Retention, and Self-Efficacy Among Moroccan Nursing Students: A Quasi-Experimental Study

**DOI:** 10.3390/healthcare13030318

**Published:** 2025-02-04

**Authors:** Hicham Blaak, Abdelmajid Lkoul, Hayat Iziki, Abderrahman Arechkik, Karim Sbai Idrissi, Samia El Hilali, Rachid Razine, Lahcen Belyamani, Ahmed Kharabch, Majdouline Obtel

**Affiliations:** 1Laboratory of Biostatistics, Clinical Research and Epidemiology, Department of Public Health, Faculty of Medicine and Pharmacy, University Mohammed V, Rabat 10100, Morocco; lkoulabdelmajid@gmail.com (A.L.); abderrahman_arechkik@um5.ac.ma (A.A.); razinerachid@yahoo.fr (R.R.); kharbach.a.lbrce@gmail.com (A.K.); majdobtel7@gmail.com (M.O.); 2High Institute of Nursing Professions and Technical Health, Agadir 80000, Morocco; izikihayat@gmail.com; 3Laboratory of Mother-Child Health and Nutrition Research, Faculty of Medicine and Pharmacy of Rabat, Mohammed V University, Rabat 10100, Morocco; 4Laboratory of Community Health, Preventive Medicine and Hygiene, Department of Public Health, Faculty of Medicine and Pharmacy, University Mohammed V, Rabat 10100, Morocco; siksbai@hotmail.com; 5Pedagogy and Research Unit of Public Health, Department of Public Health, Faculty of Medicine and Pharmacy, University Mohammed V, Rabat 10100, Morocco; samia.elhilali123@gmail.com; 6Faculty of Medicine and Pharmacy, University Mohammed V, Rabat 10100, Morocco; belyamani@gmail.com; 7Department of Medical Biotechnology and Health Sciences, University Mohammed VI of Health and Sciences, Mohammed VI Foundation of Health and Sciences, Casablanca 82403, Morocco; 8Laboratory of Cell Biology and Molecular Genetics, Department of Biology, Faculty of Sciences, Ibn Zohr University, Agadir 80000, Morocco

**Keywords:** simulation, standardized patient, knowledge acquisition, knowledge retention, self-efficacy, nursing students

## Abstract

**Background:** Simulation is a crucial component in the training of healthcare professionals, offering a safe environment for practical learning without posing risks to patients. **Aim:** This study aims to evaluate the effect of simulation with a standardized patient on the acquisition and retention of knowledge, as well as on the self-efficacy of nursing students. **Methods:** A quasi-experimental study with a pre-test and post-test design was conducted with 120 nursing students, who were randomly assigned to two groups. The experimental group (60) received training through simulation with a standardized patient, supplemented by a lecture-based course. The control group (60) received only the same lecture-based course. A pre-test was conducted to assess students’ baseline knowledge and self-efficacy before any intervention. Student performance was then assessed with a post-test immediately after the training to measure knowledge acquisition and self-efficacy and a post-test three months after the intervention to measure long-term knowledge retention. **Results:** The results showed a statistically significant difference in mean scores between the intervention and control groups in terms of knowledge acquisition (14.92 vs. 13.32; *p* < 0.001; d = 0.99), self-efficacy (33.02 vs. 31.05; *p* < 0.001; d = 1.03) and knowledge retention three months after training (12.4 vs. 10.6; *p* < 0.001; d = 0.94). **Conclusions:** The results of this study confirmed the effectiveness of simulation in knowledge acquisition and retention, as well as in the self-efficacy of nursing students. These findings support the integration of this method into training programs to strengthen the skills of future healthcare professionals.

## 1. Introduction

Nursing education today is influenced by technological, social, cultural, and economic transformations, as well as by the increasing the complexity of care required by patients [1]. These transformations emphasize the need to train healthcare professionals capable of adapting to increasingly complex clinical environments. However, this ambition is often hindered by various challenges: inadequate clinical environments, a shortage of clinical faculty, reduced internship hours, patient safety issues, and the limited availability of patients due to shorter hospital stays [2,3,4]. These challenges widen the gap between classroom-taught theory and real-world practice, making it difficult to retain clinical knowledge and skills.

In response to these constraints, simulation-based training has emerged as an innovative method to bridge this gap [5]. Simulation, defined as a process of recreating or representing a real-life situation, enables learners to develop and refine both technical and non-technical skills in a safe and controlled environment [6]. Various types of simulations are used today, including high-fidelity mannequins, virtual reality environments, hybrid simulations, and standardized patients (SPs) [7]. SPs, trained to realistically and reproducibly portray a patient, provide interactive and immersive experiences that enhance history-taking, physical examination, and communication skills [8]. They are particularly useful in psychosocial contexts requiring human and emotional interaction [9].

The effectiveness of simulation-based training is grounded in solid theoretical foundations, such as Kolb’s experiential learning theory, rooted in constructivism and postulating that learning results from the transformation of experience [10]. Kolb’s learning cycle consists of four phases: (a) concrete experience, where the learner participates in an experience like a simulation; (b) reflective observation, where the learner reflects on the experience; (c) abstract conceptualization, where the learner analyzes thoughts and reflections to identify the significance of the learning experience and considers what could have been performed differently to improve the outcome; and (d) active experimentation, which involves applying what has been learned to guide future practices [11]. In this context, SPs offer an immersive and realistic concrete experience that stimulates reflection and the development of clinical skills.

Furthermore, simulation allows for the targeted repetition of scenarios until skill mastery, even for situations rarely encountered during clinical internships [5]. This approach enhances self-confidence, knowledge, skills, critical thinking, and student satisfaction [12]. Skills acquired through simulation effectively transfer to clinical practice, strengthening overall performance, clinical judgment, and decision-making [13]. By offering a structured and secure learning environment, simulation helps learners identify their gaps, improve self-efficacy, and achieve sustainable learning outcomes [14]. However, although the benefits of simulation are widely studied, its impact on knowledge retention remains underexplored [15]. Therefore, this study aims to evaluate the effect of simulation on knowledge acquisition and retention among nursing students at the Higher Institute of Nursing Professions and Health Techniques in Agadir, as well as on their self-efficacy.

The choice of a hemorrhagic shock scenario in this study is justified by its clinical relevance as a frequent and critical emergency requiring rapid and multidimensional management. Studies show that uncontrolled hemorrhagic shock is one of the leading causes of death among trauma patients [16,17]. The early diagnosis of this condition is essential to prevent hemodynamic collapse and improve patient outcomes [18]. In the Moroccan context, where resources may be limited and certain emergency situations are rarely encountered during clinical internships, this choice ensures realistic and reproducible preparation to strengthen students’ confidence and skills.

## 2. Materials and Methods

### 2.1. Study Design and Setting

A quasi-experimental design was used in this study, with a pre-test and post-test to assess the effect of simulation on knowledge acquisition, retention, and self-efficacy among nursing students. In order to minimize selection bias and ensure initial comparability between the two groups, participants were randomly selected and then randomly assigned to either the experimental group or the control group.

The research was conducted at the Higher Institute of Nursing and Health Techniques (HINHT) in Agadir, Morocco, which is part of the higher education institutions not affiliated with universities, created by Decree No. 2.13.658 dated 30 September 2013. These institutions are under the authority of the Ministry of Health of the Moroccan government [19]. The institute includes several classrooms for theoretical courses and a simulation laboratory equipped with essential equipment (e.g., dressing sets, adult and pediatric mannequins, computers, cardiac monitors, and other essential equipment) used to enhance the clinical skills of nursing students.

### 2.2. Study Population

The study was conducted with students from the HINHT of Agadir. The inclusion criteria included enrollment in the third semester of the nursing program and agreement to participate in the study. Students who had already received training or taken a course related to the management of hemorrhagic shock in another setting, whether academic or clinical, were excluded from the study.

### 2.3. Study Variables

The independent variable of this study is the teaching method, comparing simulation with a standardized patient to a lecture-based course. The simulation involved interaction with a standardized patient, while the lecture was delivered in the form of a PowerPoint presentation on the topic of hemorrhagic shock management. The dependent variables of the study are knowledge acquisition, knowledge retention, and students’ self-efficacy.

### 2.4. Data Collection Procedure

The procedure was carried out in three stages (Figure 1): the initial phase, the acquisition phase, and the retention phase.

#### 2.4.1. Initial Phase

During this phase, students who agreed to participate in the study completed a questionnaire to collect their sociodemographic information. Then, participants from both groups took a pretest on hemorrhagic shock management knowledge and a self-efficacy questionnaire.

The pretest, teaching, and posttest were conducted sequentially. In each case, the teaching was delivered by the same instructor. All data were collected and analyzed by the principal investigator.

Students who completed the pretest were unaware of their group assignment. Students were informed of their group before the teaching began so that they did not know each other’s group membership. Each student in both groups received an identification code to anonymize their participation and protect the confidentiality of their data. There was no interaction between students in the experimental group and those in the control group.

#### 2.4.2. Acquisition Phase

During this phase, students in the control group received a lecture-based course in the form of a PowerPoint presentation on the management of hemorrhagic shock, without the opportunity to apply this intervention directly to a patient, as was possible for the experimental group. Course content included general definitions of hemorrhagic shock, its various etiologies (traumatic, surgical, obstetric, etc.), characteristic clinical signs, steps in management, as well as possible complications and interventions to prevent them. In total, 60 students participated in this session, which lasted approximately 3 h.

The experimental group, on the other hand, received the same PowerPoint presentation and then participated in a simulation scenario with a standardized patient lasting approximately 20 min (Appendix A). The 60 students who participated in the simulation experience were divided into groups of 10 for the demonstration.

An initial briefing was conducted to present the different stages of the simulation to the students, remind them of the session’s objectives, and allow them to familiarize themselves with the environment and technical equipment provided.

The simulated scenario involved a 47-year-old patient admitted to the emergency department after a high fall, presenting with a significant open wound on the left leg and active bleeding. A few minutes after admission, the patient exhibited signs indicative of hemorrhagic shock: severe hypotension (85/50 mmHg), tachycardia with a weak pulse (130 bpm), pallor, cold sweats, prolonged capillary refill time (>3 s), and mild confusion. The patient also complained of dizziness and generalized weakness.

The main objective for the students was to quickly recognize the clinical signs of hemorrhagic shock, initiate appropriate emergency interventions, and ensure effective management. They were required to assess the patient’s vital signs, apply direct compression to the wound to stop the bleeding, initiate the rapid intravenous infusion of crystalloids through two peripheral access lines, and prepare for a blood transfusion. Simultaneously, they were tasked with administering high-flow oxygen (15 L/min) via a high-concentration mask to improve tissue oxygenation, continuously monitor vital signs, and look for signs of decompensation. The students were also required to prepare a tourniquet to control the hemorrhage. They needed to work in teams, prioritize life-saving measures, maintain situational awareness, and communicate effectively with the emergency medical team. Finally, they had to accurately document the interventions performed and their timing and evaluate the effectiveness of the actions taken to ensure optimal continuity of care during the patient’s transfer to an emergency team or a hospital department. The simulation sessions were not recorded due to the absence of a functional recording system suitable for this purpose.

A debriefing session, lasting approximately 40 min, was conducted at the end of the simulation training to discuss and clarify points raised during the scenario. This debriefing was structured into three phases: descriptive, analytical, and applicative. In the descriptive phase, the students recounted the experienced situation, expressed their feelings, and identified the main challenges encountered. The analytical phase focused on examining the decisions made, assessing successful or improvable actions, and exploring the underlying reasons. Finally, the applicative phase encouraged participants to propose avenues for improvement, reflect on optimizing their practices, and consider concrete ways to integrate the knowledge gained into future situations.

Once both sessions were completed, the first post-tests, including the general self-efficacy scale and the knowledge assessment, were administered to evaluate the students’ knowledge and their perception of self-efficacy.

#### 2.4.3. Retention Phase

Three months after the intervention, students from both groups were invited to participate in the knowledge retention phase, which involved retaking the knowledge retention test using the same pretest questionnaire but with the items reordered.

### 2.5. Study Materials

The course materials were designed to ensure consistency with the learning objectives and optimize student learning. The theoretical courses were delivered in a classroom equipped with computers and projectors, enabling an interactive and dynamic presentation. PowerPoint presentations were the primary tool for lectures, providing clear and concise definitions of hemorrhagic shock, its etiologies, clinical signs, and management protocols in accordance with international standards (Appendix A). To illustrate these theoretical concepts, algorithms and explanatory diagrams were incorporated. Additionally, printed explanatory documents were distributed to learners. For practical simulations, the available equipment created an authentic hospital-like environment. The emergency simulation room was equipped with technical devices such as cardiac monitors, intravenous administration systems, and high-concentration oxygen therapy masks. A well-stocked emergency cart was also present, containing essential medications for emergency care, including intravenous crystalloids (0.9% NaCl) for fluid resuscitation, blood products (packed red blood cells, fresh frozen plasma) to compensate for massive losses, vasoactive drugs (dopamine, norepinephrine) for refractory shock, antifibrinolytics (tranexamic acid) to limit bleeding, as well as analgesics like morphine for pain management. The cart also contained essential materials such as infusion sets, catheters, oxygen therapy masks, tourniquets, and syringes. This equipment and supplies enabled the creation of realistic scenarios to prepare students for emergency situations.

### 2.6. Standardized Patient

In this study, the SP is defined as a person trained to simulate a specific medical case by reproducing its clinical signs, personality, body language, and emotions. The simulated patients selected for this study were experienced individuals trained by simulation experts. During the preparation for the scenario, the standardized patients were briefed on the nature of their role, the scenario steps, as well as the objective and procedure of the simulation. The SPs worked according to structured guidelines and performed a minimum of five case trials before the simulation. These trials were supervised by the program facilitators, who made minor adjustments to ensure optimal standardization and provide a comparable experience for each student. The SP was monitored using a simulation monitor, utilizing the free online version of the Vital Sign Simulator software (available at https://sourceforge.net/projects/vitalsignsim/, accessed on 26 October 2024). This software allowed for the control of hemodynamic parameters and their adjustment based on the scenario’s progression.

### 2.7. Measurement

Knowledge test: The knowledge assessment tool consists of a 20-item multiple-choice questionnaire on the management of hemorrhagic shock (Appendix A). This questionnaire was developed based on specific educational objectives, including the rapid recognition of clinical signs of hemorrhagic shock and the initiation of appropriate emergency interventions, ensuring adequate tissue oxygenation, continuous monitoring of vital signs and clinical status, and effective coordination of care with the emergency medical team. The questionnaire was developed by the researchers and then reviewed and validated by a group of expert teachers in the field. Before distribution, it was tested on 30 students who were excluded from the main study to evaluate the reliability and clarity of the questions and instructions, as well as the time required to complete it. Minor modifications were made based on written feedback from the participants. The final version of the questionnaire includes 20 questions, with a Cronbach’s alpha reliability coefficient of 0.85.

Knowledge scores are calculated based on correct or incorrect answers, ranging from 0 to 20, with higher scores reflecting a higher level of knowledge.

General self-efficacy scale (GSE): The general self-efficacy scale, or the Personal Efficacy Scale, was initially developed by Matthias Jerusalem and Ralf Schwarzer in 1981 [11]. The scale consists of 10 items, each rated on a Likert scale from “1” (not at all true) to “4” (exactly true). The total score ranges from 10 (minimum self-efficacy) to 40 (maximum generalized self-efficacy). This scale has been translated and validated in French by Dumont and his team (α = 0.82) [20,21] (Appendix A).

### 2.8. Bias

To minimize biases in this study, a single instructor conducted all teaching sessions, following standardized training prior to the study’s initiation. This training included a comprehensive review of teaching protocols and study objectives. Detailed instructional scripts were utilized during all sessions to ensure consistency in content delivery and interactions across groups. Identical educational materials were provided to all participants, and random recordings of sessions were reviewed to verify compliance with the established protocols. To control for bias in the evaluation process, self-administered questionnaires were used, ensuring that participant responses were not influenced by the instructor. These measures were implemented to reduce potential instructor-related and evaluation-related biases while maintaining uniformity between the experimental and control groups.

### 2.9. Data Analysis

The data were analyzed using the “Statistical Package for the Social Sciences” (SPSS 27 developed by IBM Corporation, New York, NY, USA). Descriptive statistics (frequencies, percentages, means, and standard deviations) were used according to the measurement level to describe the sample and study variables. An independent Student’s *t*-test was then applied to compare the mean differences in the study variables (knowledge acquisition, knowledge retention, and self-efficacy) between the experimental group and the control group. Additionally, a paired *t*-test was used for comparing results within the same group.

The sample size of 120 students (60 per group) was determined through a power analysis aimed at detecting significant differences in knowledge retention between the two groups. The analysis was based on an anticipated medium effect size (0.5), with a power of 80% and a significance level of 0.05.

## 3. Results

### 3.1. Participants’ Characteristics

In our study, all 120 initially enrolled nursing students completed the demographic data form and successfully participated in all phases of the study (Table 1). The mean age of the intervention group (IG) was 20.22 years (±1.71), slightly higher than that of the control group (CG), which was 19.93 years (±1.16). The IG had a higher percentage of female students (56.4%) compared to the CG (43.6%), while the CG had a larger proportion of male students (61.9% vs. 38.1%). The majority of participants were single. The IG had a slightly higher number of participants from rural areas (56.3% vs. 43.8%). Income levels were similar between the groups. There were also no notable differences in residence types. Most participants had passed the semester. A larger number of students in the CG had prior simulation training (57.1% vs. 42.9%). Overall, the baseline characteristics of the two groups were largely comparable, with no statistically significant differences.

An independent *t*-test showed no significant difference between the intervention and control groups regarding knowledge (Mean ± SD: 7.42 ± 3.17 vs. 8.08 ± 2.69, *p* = 0.217) or self-efficacy (Mean ± SD: 25.0 ± 4.16 vs. 24.9 ± 4.24, *p* = 0.862). These results indicate that the groups were well matched at the start of the study, allowing for the attribution of observed effects to the intervention rather than to differences between the groups.

### 3.2. Intra-Group Comparisons of Knowledge and Self-Efficacy

As shown in Table 2, in the IG, the mean knowledge score significantly increased from 7.42 (±3.17) at the pre-test to 14.92 (±1.66) at the post-test (*t* = 21.65, *p* < 0.001). Similarly, the CG also showed a significant increase in knowledge scores, from 8.08 (±2.69) at the pre-test to 13.32 (±1.57) at the post-test (*t* = 20.07, *p* < 0.001).

The IG also demonstrated a significant improvement in self-efficacy, with scores rising from 25.03 (±4.16) at the pre-test to 33.02 (±1.44) at the post-test (*t* = 15.39, *p* < 0.001). The CG also experienced an increase in self-efficacy scores, from 24.90 (±4.24) at the pre-test to 31.05 (±2.29) at the post-test (*t* = 15.69, *p* < 0.001). These results indicate that both the intervention and control groups experienced significant improvements in knowledge and self-efficacy following the training.

### 3.3. Inter-Group Comparisons of Knowledge, Self-Efficacy, and Knowledge Retention

The results of this study show significant differences between the IG and CG in terms of knowledge acquisition and self-efficacy. Participants in the IG achieved a higher mean knowledge score (M = 14.92, SD = 1.66) compared to the CG (M = 13.32, SD = 1.57). This difference was statistically significant (*t* = 5.43, *p* < 0.001) with a large effect size (Cohen’s d = 0.99). The confidence interval for the effect size ranged from 0.61 to 1.37, indicating a strong and meaningful impact of the intervention on knowledge improvement.

For self-efficacy, the IG also reported higher scores (M = 33.02, SD = 1.44) than the CG (M = 31.05, SD = 2.29). This difference was statistically significant (*t* = 5.63, *p* < 0.001), with a large effect size of Cohen’s d = 1.03. The confidence interval for this effect size was between 0.60 and 1.41, further supporting the significant advantage of the intervention.

Three months later, during the retention phase, a post-test was conducted to assess knowledge retention. The results again showed a statistically significant difference between the IG (M = 12.4, SD = 2.01) and the CG (M = 10.6, SD = 1.76) (*t* = 5.13, *p* < 0.001). The effect size for this difference was large (Cohen’s d = 0.94), with a confidence interval ranging from 0.56 to 1.31. As presented in Table 3, these findings demonstrate that the intervention had a substantial and lasting impact on knowledge retention.

## 4. Discussion

This study examined the effectiveness of simulation-based training with standardized patients on knowledge acquisition, retention, and self-efficacy among nursing students in Morocco. The results demonstrate that simulation is a powerful educational tool that not only enhances immediate knowledge acquisition but also supports long-term retention and the development of self-efficacy, with significant effect sizes observed across all outcomes.

Participants in the simulation group achieved significantly higher post-test scores (14.92 ± 1.66) compared to the control group (13.32 ± 1.57; *p* < 0.001; d = 0.99). This substantial effect highlights the educational value of integrating simulation into nursing education programs. These findings align with the existing literature that emphasizes the effectiveness of simulation-based learning in improving educational outcomes in nursing. For instance, a meta-analysis by Oh et al. [22] showed that simulation-based learning with standardized patients significantly enhances knowledge acquisition compared to traditional teaching methods (d = 0.38, *p* = 0.05, I^2^ = 42%). Similarly, Tubaishat and Tawalbeh [23] reported notable improvements in students’ knowledge scores after simulation-based training on cardiac arrhythmias, with large effect sizes (13.2 ± 3.35 vs. 7.6 ± 2.36; *p* < 0.001; d = 1.92). Becker et al. [24] observed moderate improvements in communication skills and knowledge among final-year nursing students participating in a 7-week psychiatric care simulation (d = 0.36).

Empirical studies confirm that simulation is an effective and adaptable educational tool in nursing [14,25,26,27,28,29,30]. This study stands out for its integration of standardized patients, bringing realism and promoting communication and decision-making under pressure. Unlike lectures, this approach focuses on active and interactive learning, enabling students to translate theoretical knowledge into practical skills. By fostering immersion and direct participation, simulation fully engages students, thereby enhancing their performance in real clinical contexts [31]. Furthermore, it is well established that students’ attention significantly decreases after 15 to 20 min of traditional teaching, limiting the effectiveness of lectures [32].

By replicating real-life situations, simulation places students in contexts where they can act as they would in a clinical setting [33]. It provides a safe and controlled environment where learners can practice their clinical knowledge and skills, learn from their mistakes, and make decisions without compromising patient safety [34,35]. Through error analysis and feedback provided by instructors, students strengthen their skills and enhance knowledge retention and application [36]. Additionally, simulated scenarios promote an experiential and interactive learning environment, enabling the construction of conceptual frameworks and fostering students’ intellectual and emotional engagement [37,38]. The interactivity and authenticity of simulation sharply contrast with the limitations of traditional pedagogy, where the absence of such elements may reduce learners’ motivation and engagement. Furthermore, the immersion offered by simulation in realistic contexts stimulates multiple learning modes, including visual, auditory, and kinesthetic, thus engaging students more deeply.

Unlike the often superficial responses observed during lectures, simulation encourages in-depth reflection and active participation. In this study, each scenario was followed by debriefing sessions, a crucial element of this teaching method. These sessions allowed for the analysis and correction of errors, enhancing knowledge assimilation. This approach aligns with the findings of other studies highlighting that debriefing improves knowledge retention, corrects inappropriate behaviors, and boosts students’ confidence [33]. By offering a space for interaction, it allows learners to ask questions, receive constructive feedback, and share their perceptions. This consolidates learning and strengthens self-confidence, significantly improving overall competencies. Kiernan demonstrated that deliberate practice and video debriefing were effective simulation modalities for acquiring nursing skills and self-assessment [39].

Retention scores, three months after the intervention, were significantly higher in the simulation group (12.4 ± 2.01) compared to the lecture group (10.6 ± 1.76; *p* < 0.001; d = 0.94). The large effect size observed in this study reinforces the utility of simulation in maintaining long-term knowledge retention. This is particularly important for hemorrhagic shock, where rapid and accurate decision-making is critical. This finding aligns with the conclusions of Araújo et al. [40] and Costa et al. [41], who also reported better long-term knowledge retention following simulation-based training, with effect sizes of (d = 0.78 and d = 0.67), respectively. These benefits are attributed to active engagement and contextual learning, which enhance memory consolidation. The immersive and interactive aspects of simulation promote durable memorization by stimulating emotional and cognitive engagement. Other studies support these results, highlighting the advantages of simulation in addressing skill gaps often observed in traditional education. A randomized controlled trial demonstrated that virtual reality simulation significantly improved the retention of ECG interpretation skills, surpassing traditional teaching methods [42]. These approaches foster self-directed learning while providing realistic clinical scenarios, strengthening the practical application of theoretical knowledge [43,44]. These findings underscore the potential of simulation-based methodologies to ensure the durability of learning.

The simulation group also achieved significantly higher self-efficacy scores (33.02 ± 1.44) compared to the lecture group (31.05 ± 2.29; *p* < 0.001; d = 1.03). This notable effect highlights the effectiveness of simulation-based education in enhancing the self-efficacy of nursing students. High self-efficacy is a critical outcome as it influences students’ confidence in clinical decision-making and their ability to apply knowledge in high-pressure environments. These findings are consistent with previous studies. Al Gharibi et al. [45] observed a significant improvement in perceived confidence and skills among students trained through simulation in emergency management (d = 0.92). Similarly, Khadivzadeh and Erfanian [46] reported a moderate improvement in self-efficacy and a reduction in anxiety among midwifery students following practical simulation (d = 0.47). Other works corroborate these observations, showing that simulation enhances self-efficacy, critical thinking, perceived learning, and students’ skills [47,48,49].

These beneficial effects can be attributed to the very nature of simulation, which promotes active learning by immersing students in authentic environments where they must make real-time decisions, solve complex problems, and engage with dynamic scenarios [6]. By working in teams and within a collaborative environment, students develop critical thinking skills, broaden their perspectives, increase their motivation, and identify both their strengths and areas for improvement [50]. This process engages cognitive, emotional, and psychomotor dimensions, thereby enhancing their ability to integrate and apply knowledge holistically. Furthermore, the experiential learning offered by simulation establishes a strong connection between theory and practice, consolidating the understanding of the concepts addressed [51]. Active learning also contributes to the development of clinical reasoning by placing students in complex situations that require in-depth analysis, informed decision-making, and the prioritization of care.

The findings of this study suggest that simulation-based training is an effective educational strategy, even in resource-constrained environments such as Morocco. The substantial improvements in knowledge acquisition and retention observed in this study underscore the potential for targeted investments in simulation infrastructure and instructor training to produce meaningful educational outcomes, particularly in regions where clinical exposure opportunities are limited. Future research should explore the cost-effectiveness and practical feasibility of implementing simulation-based training in similar contexts.

Moreover, these results highlight the critical value of simulation in bridging the gap between theoretical knowledge and clinical application, specifically for hemorrhagic shock management. By fostering active and experiential learning, simulation enables students to deeply engage with the material and develop the practical skills essential for real-world clinical settings. These findings are particularly relevant for resource-limited settings like Morocco, where simulation-based training can serve as a valuable tool to mitigate the challenges posed by restricted access to clinical exposure and enhance the quality of nursing education.

### Limitations

This study has several limitations that must be acknowledged. First, the single-site design and the inclusion of only second-year undergraduate nursing students limit the generalizability of the findings. Future research should include multi-center studies with diverse student populations to enhance external validity. Additionally, the relatively short follow-up period precludes a thorough understanding of long-term knowledge retention. Incorporating longer follow-up intervals in future studies would provide deeper insights into the durability of simulation-based learning outcomes.

Another limitation of this study is its focus solely on cognitive knowledge, without examining how this knowledge is applied in clinical practice. To address this gap, future research should incorporate rigorous evaluation tools, such as objective structured clinical examinations, to assess the effectiveness of simulation-based training in developing practical skills and clinical competencies. Such methods would offer valuable insights into the real-world impact of this training on healthcare delivery.

Finally, the absence of recorded debriefing sessions deprived students of opportunities for self-review and error analysis. Including multimedia tools for recording and reviewing simulation exercises could enrich the learning experience and foster deeper reflection in future iterations of this research.

## 5. Conclusions

The results of this study highlighted the effectiveness of simulation in knowledge acquisition and retention, as well as in enhancing the self-efficacy of nursing students. These data suggest that integrating simulation into training programs could address challenges faced by nursing education, such as increasing class sizes, a lack of clinical facilities for placements, and a shortage of instructors. While the results are promising, further research is needed to determine the optimal frequency of simulation sessions and to assess the long-term impact on knowledge and skill retention. Therefore, it is recommended that policymakers and educators adopt this pedagogical approach to better prepare students for the current clinical demands.

## Figures and Tables

**Figure 1 healthcare-13-00318-f001:**
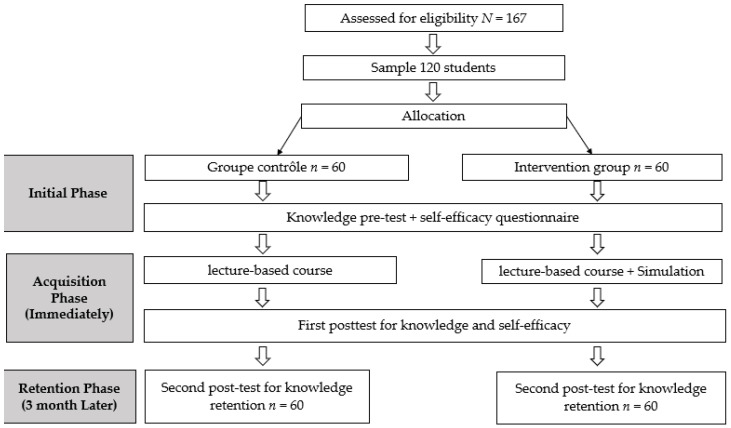
Study flow diagram.

**Table 1 healthcare-13-00318-t001:** Baseline characteristics of participants in the intervention and control groups.

Variables		Intervention Group	Control Group	*p*-Value
Age (Mean ± SD)		20.22 ± 1.708	19.93 ± 1.163	0.290
		*N* (%)	*N* (%)	
Gender	Female	44 (56.4)	34 (43.6)	0.056
	Male	16 (38.1)	26 (61.9)	
Marital status	Single	58 (49.2)	60 (50.8)	0.154
	Married	2 (100)	0 (0)	
Origin	Rural	18 (56.3)	14 (43.8)	0.409
	Urban	42 (47.7)	46 (52.3)	
Income level	High	5 (41.7)	7 (58.3)	0.825
	Low	13 (52.0)	12 (48.0)	
	Medium	42 (50.6)	41 (49.4)	
Residence type	With Family	37 (49.3)	38 (50.7)	0.312
	Shared housing	20 (57.1)	15 (42.9)	
	University residence	3 (30.0)	7 (70.0)	
Semester validation status	No	1 (33.3)	2 (66.7)	0.559
	Yes	59 (50.4)	58 (49.6)	
Prior simulation training	No	33 (57.9)	24 (42.1)	0.100
	Yes	27 (42.9)	36 (57.1)	
Knowledge of hemorrhagic shock (Mean ± SD)		7.42 ± 3.17	8.08 ± 2.69	0.217
Self-efficacy (Mean ± SD)		25.0 ± 4.16	24.9 ± 4.24	0.862

SD: standard deviation.

**Table 2 healthcare-13-00318-t002:** Paired *t*-test to compare knowledge and self-efficacy in the pre- and post-test within groups.

Variables		Pretest (Mean ± SD)	Post-Test (Mean ± SD)	*t*-Test	*p*-Value
Knowledge of hemorrhagic shock	IG	7.42 ± 3.17	14.92 ± 1.66	21.65	<0.001
	CG	8.08 ± 2.69	13.32 ± 1.57	20.07	<0.001
Self-efficacy	IG	25.03 ± 4.16	33.02 ± 1.44	15.39	<0.001
	CG	24.90 ± 4.24	31.05 ± 2.29	15.69	<0.001

IG: intervention group, CG: control group.

**Table 3 healthcare-13-00318-t003:** Difference in knowledge, self-efficacy, and knowledge retention between groups.

Variables	IG (Mean ± SD)	CG (Mean ± SD)	*t*-Test	*p*-Value	Cohen’s d [CI 95%]
Knowledge acquisition	14.92 ± 1.66	13.32 ± 1.57	5.43	<0.001	0.99 [0.61–1.37]
Self-efficacy	33.02 ± 1.44	31.05 ± 2.29	5.63	<0.001	1.03 [0.6–1.41]
knowledge retention	12.4 ± 2.01	10.6 ± 1.76	5.13	<0.001	0.94 [0.56–1.31]

IG: intervention group, CG: control group, CI: confidence interval, SD: standard deviation.

## Data Availability

All data generated or analyzed during this study are included in this published article.

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
