# Peer review of "Effectiveness of Simulation with a Standardized Patient on Knowledge Acquisition, Knowledge Retention, and Self-Efficacy Among Moroccan Nursing Students: A Quasi-Experimental Study"

_healthcare, 2025, doi:10.3390/healthcare13030318_

Round 1
Reviewer 1 Report
Comments and Suggestions for Authors
The paper entitled “Effectiveness of simulation with standardized patients on knowledge acquisition, knowledge retention, and self-efficacy among Moroccan nursing students: A quasi-experimental study," which presents a quasi-experimental study with 120 nursing students, is interesting. However, I have many concerns about this study:
First, the authors present a topic without providing a theoretical background on simulation. There are many theoretical foundations [e.g., Cognitive Theory of Multimedia Learning (Mayer, 2002)] proposing design guidelines for simulations. Moreover, many empirical studies focused on simulations for educational reasons and especially in healthcare, etc. (Elendu et al., 2024; Hippe et al., 2020). The authors present a shallow introduction without providing thorough related work on this topic.
Second, the methodology has many flaws in several sections. There is little information about the topic materials. Some screenshots of the simulation materials would be helpful. Also, the didactic strategy is not known. I would recommend the authors write a subsection regarding the study materials. Similarly, an appendix with extra details about the study materials would be necessary to follow.
Regarding the measures, the authors could provide some examples of the knowledge test items.
Third, the discussion has no flow. The iteration of the word “simulation” in the first line of each paragraph is somewhat boring. Moreover, the authors cite some relevant studies to relate their results without mentioning them in the first section of their paper.
I would recommend the authors revise and resubmit their paper.
References
Elendu, C., Amaechi, D. C., Okatta, A. U., Amaechi, E. C., Elendu, T. C., Ezeh, C. P., & Elendu, I. D. (2024). The impact of simulation-based training in medical education: A review. Medicine, 103(27), e38813.
Hippe, D. S., Umoren, R. A., McGee, A., Bucher, S. L., & Bresnahan, B. W. (2020). A targeted systematic review of cost analyses for implementation of simulation-based education in healthcare. SAGE Open Medicine, 8, 2050312120913451.
Mayer RE (2002) Multimedia learning. In psychology of learning and motivation (Vol. 41). Academic Press, pp. 85–139.
Author Response
Comment 1: Insufficient theoretical context: The authors present a topic without providing a theoretical context on simulation. There are numerous theoretical frameworks [e.g., Mayer’s Cognitive Theory of Multimedia Learning (2002)] offering guidelines for designing simulations. Additionally, many empirical studies focus on simulations for educational purposes, especially in healthcare (Elendu et al., 2024; Hippe et al., 2020).
Response:
We sincerely thank you for your valuable remarks and constructive comments, which have greatly enhanced the quality of our manuscript. The theoretical framework you mentioned was particularly helpful and highly relevant. Significant changes have been made to the introduction, including a detailed theoretical background that incorporates Kolb's experiential learning theory and recent empirical references on the use of simulations in healthcare, particularly Elendu et al. (2024). These additions are found in the second and third parts of the introduction (page 2; lines 52–71).
Comment 2: Superficial introduction: The authors present a superficial introduction without including in-depth studies on this topic.
Response:
The introduction has been enriched with a more comprehensive review of recent studies regarding the use of simulations for educational purposes, with a particular focus on the healthcare field. This section is now more detailed (introduction 43-93)
Comment 3: Insufficient information on teaching materials: There is little information on the teaching materials. Screenshots of the simulation materials would be helpful. Additionally, the teaching strategy is unclear. I recommend the authors add a subsection about the study materials and provide an appendix with additional details.
Response:
Teaching Materials and Didactic Strategy has been added in the Methodology section (page 5; 195-213). Details on the materials used and their design are included. Additionally, screenshots of the simulation materials have been provided in the Appendix for better understanding.
Comment 4: Measures: Provide examples of items from the knowledge test.
Response:
Examples of items from the knowledge test have been added in the measurement section (page 5; 231-235). The knowledge test has been provided in the appendix for better understanding.
Comment 5: Lack of fluency in the discussion: The repeated use of the word “simulation” at the beginning of each paragraph is somewhat tedious.
Response:
The discussion has been revised to improve fluency and avoid excessive repetition of the word simulation (discussion part, page 8-10)
Comment 6: Studies cited in the discussion but missing in the introduction: Relevant studies cited in the discussion are not mentioned in the introduction.
Response:
The relevant studies mentioned in the discussion have been integrated into the introduction to ensure coherence throughout the article. (page 9; 361; 426 )
Reviewer 2 Report
Comments and Suggestions for Authors
Hicham Blaak et al. presented a study entitled: "Effectiveness of simulation with standardized patients on knowledge acquisition, knowledge retention, and self-efficacy among Moroccan nursing students: A quasi-experimental study" which aimed to address a critical gap in nursing education by exploring the effectiveness of simulation with standardized patients.
The research topic is timely and well-aligned with current educational challenges, especially in nursing schools. However, several issues need to be addressed to stress out the importance and significance of the current paper.
(1) In terms of Relevance and clarity, can the authors explain why the hemorrhagic shock as the simulation focus was prioritized over others?
(2) Methodology:
(2.1) Please provide a more detailed description of the simulation scenario used in this study, including specific learning objectives, the role of the standardized patient, and the debriefing process.
(2.2) The manuscript lacks detail on how the study controlled for potential instructor bias during the teaching and evaluation phases. Were any measures taken to ensure uniformity across sessions?
(2.3) Please clarify the rationale for not recording the debriefing sessions
(3) Results: The data presentation section could be improved. For instance some tables (e.g., Table 1) could benefit from more descriptive titles and captions to enhance readability. Additionally, while Cohen's d values are reported, including confidence intervals in the main text would provide a clearer understanding of the intervention's effect size.
(4) Discussion section: Although the discussion section thoroughly contextualizes the findings within existing literature, but it occasionally reiterates the results without deeper analysis.
(4.1) Could you explore the implications of the effect sizes observed in your study for scaling simulation-based training in resource-constrained settings like Morocco?
(4.2) Consider discussing how simulation promotes active learning, experiential learning and the development of clinical reasoning skills.
(5) Study limitation.
(5.1) The current section is ok, but it overlooks the potential impact of the short follow-up period on knowledge retention outcomes. Have you considered including a longer-term retention assessment in future studies?
(5.2) Can you elaborate on the need for future research to explore the impact of such scenario on clinical performance using more robust assessment methods?
(5.3) Please expand to discuss the generalizability of the findings given the single-site study design and the specific characteristics of the student population.
6. Language and style. The paper could benefit of proofreading as it has occasional lapses in fluency (e.g., “Simulation promotes active learning and allows students…”). Please take time to consider revising for smoother transitions and conciseness.
The manuscript is written in clear academic English. Nevertheless, it requires proofreading to address these occasional lapses in fluency, and typos errors.
Author Response
Comment 1: Why prioritize hemorrhagic shock as a simulation objective?
Response:
We sincerely thank you for your valuable remarks and constructive comments, which have greatly enhanced the quality of our manuscript
A detailed justification has been added to the introduction (page 2, 83-90) explaining why hemorrhagic shock was prioritized. This objective was chosen due to its critical clinical importance and the necessity for nurses to be trained to effectively manage life-threatening situations.
Comment 2: Detailed description of the simulation scenario: Please provide specific learning objectives, the role of the standardized patient, and the debriefing process.
Response:
A comprehensive description of the simulation scenario, including specific learning objectives and the steps of the debriefing process, has been added to the Methodology section (page 4, 153-184).
We have added a dedicated subsection in the Methodology section to detail the role of the standardized patient (page 5; 215)
Comment 3: Control of potential instructor bias: The manuscript lacks details on how uniformity between sessions was ensured.
Response:
A subsection addressing the control of potential biases has been added to the Methodology section (page 6; 250). We explained that all instructors underwent standardized training and that strict protocols were implemented to ensure uniformity across sessions.
Comment 3: Recording of debriefing sessions: Please clarify why debriefing sessions were not recorded.
Response:
The reason for not recording debriefing sessions has been added: The simulation sessions were not recorded due to the absence of a functional recording system suitable for this purpose (page 4,174)
Comment 4: Improving tables and adding confidence intervals: Some tables could benefit from more descriptive titles, and confidence intervals should be included.
Response:
The titles of the tables have been revised for greater clarity (see tables).
Confidence intervals have been added to the main results in the text (page 8; 320-337)
Comment 5: In-depth analysis of effect sizes for low-resource settings
Response:
An in-depth analysis of the implications of observed effect sizes for low-resource settings, such as Morocco, has been added to the discussion (page 10, 438-453). We explored how these results can be adapted to training in these contexts.
Comment 6: Simulation and active learning: Consider discussing how simulation fosters active learning, experiential learning, and the development of clinical reasoning.
Response:
the impact of simulation on active learning, experiential learning, and the development of clinical reasoning skills has been incorporated into the discussion (page 10; 425-436)
Comment 7: The limitations section overlooks the impact of the short follow-up period on knowledge retention results.
Response:
The potential impact of the short follow-up period on knowledge retention results has been mentioned in the Limitations section (page 11; 459-462).
Comment 8: Explain why future research is necessary to evaluate clinical performance.
Response:
We have highlighted the need for future studies to incorporate robust evaluation methods to provide a clearer understanding of how simulation-based training translates into practical skills and clinical competencies, ensuring its relevance and impact on real-world healthcare settings (page 11; 463-468).
Comment 9: Discuss the generalizability of results given the single-site design and specific characteristics of the population.
Response:
A discussion on the generalizability of results has been added to the Limitations section (page 11 ; 455-459). We addressed the challenges related to the single-center design and specific characteristics of the study population
Comment 10: The document could benefit from a thorough revision to improve fluency and transitions.
Response:
A complete revision of the text was conducted to improve fluency and conciseness. Transitions between sections were reworked, and long or complex sentences were simplified.
Reviewer 3 Report
Comments and Suggestions for Authors
First of all I want to thank you for your effort. It would be more appropriate to share the questions used in the survey and evaluate the results according to these questions. Because the results may vary depending on whether the questions are about basic information or practice.
Author Response
Comment 1: It would be more appropriate to share the survey questions and evaluate the results based on these questions. The results may vary depending on whether the questions address basic information or practical skills.
Response:
We sincerely thank you for your valuable remarks and constructive comments, which have greatly enhanced the quality of our manuscript.
The survey questions have been added in the appendix for better transparency and a more thorough understanding of the evaluation.
Round 2
Reviewer 1 Report
Comments and Suggestions for Authors
The authors have fixed some crucial parts of the paper. The paper has flow. Also, the authors extended the methodology providing more information regarding the measurement tools and bias.
The paper should be accepted.
Reviewer 3 Report
Comments and Suggestions for Authors
Thank you very much for your effort.